# Cell Segmentation in Multi-modality High-Resolution Microscopy Images with Cellpose

**Hyungjo Byun**†
School of Electrical and Computer Engineering
University of Seoul
163 Seoulsiripdaero, Dongdaemun-gu, Seoul, Republic of Korea
qusjo8@uos.ac.kr

**Kwanyoung Lee**†
School of Mechanical Engineering
Hanyang University
222 Wangsimni-ro, Seongdong-gu, Seoul, Republic of Korea
mobled37@hanyang.ac.kr

**Hyunjung Shim**∗
Kim Jaechul Graduate School of Artificial Intelligence
KAIST
85 Hoegi-ro, Dongdaemun-gu,Seoul 02455, Republic of Korea
kateshim@kaist.ac.kr

## Abstract

Deep learning has achieved great results in microscopy image processing in the field of Biology. However, a generalized model is needed which can solve overfitting and produce good performance for test images and unseen classes. The reason why it is difficult to make a generalized model is because of the diversity of modality, staining methods, cell shapes and resolution of microscopy images. The dataset of the "Weakly Supervised Cell Segmentation in Multi-modality High-Resolution Microscopy Images" challenge consists of images with these diverse characteristics. Therefore, we trained cellpose[1] to perform instance segmentation well on dataset having various characteristics. In order to apply the cellpose model to the challenge dataset, we designated model to always use green and blue channels for any type of images. What's more we newly created the size estimation model predicting diameter of cell to operate on various resolutions. As a result, we could achieve F1 score 0.7607 for the validation (Tuning) set.

## 1 Introduction

Recently in biology area, deep learning accomplished great progress processing microscopy image like tissue cell or bacteria[2][3]. In particular, the cell segmentation model developed under fully supervised setting shows excellent performance in training data, but as a result of overfitting by memorization, there is a significant performance degradation during test time[4]. This overfitting issue creates a bigger problem when applying the same model to the unseen category microscopy image. In other words, a model specialized in a specific category cannot be applied to an unseen

---

∗corresponding author
†Equal contribution

36th Conference on Neural Information Processing Systems (NeurIPS 2022).

classes because its generalization ability is poor. In addition, the performance is insufficient as a pretrained model for transfer learning[1]. Therefore, it is important to make general model which is applicable to variety of cells.

Many attempts have been made in the field of Cell Segmentation to create models that applicable to various types of microscopic images. However, there are two reasons that there isn't meaningful method for creating a generalized model yet. The first is that the complexity of relation between input and output is high due to the diversity of cell observations and staining methods[5]. There are various microscopic modalities such as bright field, fluorescent, phase-contrast, and differential interference contrast. Moreover, the image channels and expression formats of each modality are diverse. Even within the same modality, different staining methods can be used which can increase the complexity of the data distribution. For example, in the fluorescent image, the dye and the wavelength of the fluorescence varies depending on the protein to be displayed. In addition, the result images are generated in various ways according to cellular morphology, RNA expression, and protein expression. The second is defining relation between input and output is non-trivial due to the diversity of cell shapes. Even same type cells are difficult to separate if they are tightly packed together, and the shape of the colonial morphology varies[1]. In addition, differences in the resolution of images is additional reason that varying cell shapes. The diversity of resolution varies the boundary shape of cells and the distribution of colonies in images[6]. This diversity of cell shapes makes it difficult for deep learning models to robustly extract features.

"Weakly Supervised Cell Segmentation in Multi-modality High-Resolution Microscopy Images" challenge addresses the issues of the aforementioned cell image variation and aims to develop a generalized cell segmentation model. First, the microscopic modalities included in this Challenge's dataset is bright field, fluorescent, phase-contrast, and differential interference contrast, and each modality has a different dyeing method, image size, channels and so on. In addition, dataset contains various cells, so its shape and size are different. Finally, because the image resolution is not unified, there are images of resolution ranging from 512x512 to 10,000x10,000.

Therefore, we used cellpose[1] to deal with the difficulties caused by modality, such as dyeing and channel diversity, and the diversity of cell shapes caused by the diversity of colonial morphology and resolution. Cellpose is a U-Net-based instance segmentation model[2] that takes microscopy image as input and outputs the probability of cells and a gradient vector field that converges to the center of cells for each pixel. The reason why we chose cellpose among the existing SOTA models is that we thought Cellpose would handle the diversity of modality and cell shapes better than other models. Cellpose provides a pretrained model for each clustering according to the modality of the image during the pretraining process. Therefore, since the cellpose method shows good performance for individual modality, we thought that a general model would be possible if modalities were trained at the same time. On the other hand, there are three reasons why cellpose is suitable for solving problems caused by various cell shapes. First, unlike models that output only pixel-specific cell probabilities, cellpose uses gradient of flow to segment cells. Therefore, it works well even with slight errors in classification. Second, compared to the methods using star-convex polygon such as a stardist[3], cellpose does not assume a predetermined polygon, so it works well for elongated and complex shape of cells. Finally, it is expected that cellpose will also robust to the diverse resolution because it performs resize based on diameter estimation using style vector which encode the style of the input image.

However, two problems must be solved before the cellpose model can be used for dataset of challenge. The first problem is difference of the channel definition between the pretraining process and the challenge dataset. Cellpose receives two channels as input, the first channel showing cytoplasm and the second channel showing cell nuclei. At this time, if there is no specified channel for displaying the cell nucleus, the second channel is filled with zero. However, in the challenge's dataset, there are cells that have no nucleus and cells that cannot be distinguished without the color of the dye. Also, the number of channels varies. In this case, it is difficult to convert into two channel images defined by Cellpose. Therefore, we decided to fill the second channel with zero for images having only one channel, as proposed by cellpose, and use any two of the original image's channels for images with two or more channels. Therefore, taking advantage of the fact that there are many images in dataset that are grean cytoplasm and blue nucleus, we fixed the first channel as green and the second as blue, and leave the model to adapt the other multi channel images itself.

The second problem when applying Cellpose to the challenge dataset is the diversity of resolution. The image of challenge dataset has various resolution, and there are images that are much higher resolution than cellpose's pretraining dataset. Thus, assuming that the cell diameter is fixed to the size used during pretraining, even the same cell has different accuracy depending on the resolution[7]. Therefore, in order to handle various resolutions, the cell size expressed in the original image was predicted and resized to the cell size used during training. As a result of using the newly trained cellpose model after modifying it to suit the challenge dataset, the F1 score for validation set was 0.7607.

In the method of this report, we will explain the structure and training method of the model after mentioning why the cellpose is superior to other models. In experiments, we will mention about the characteristics of the data used in cellpose pretraining and the data covered in challenge. After that, we will explain implementation details. Finally, final results ,visualization of the results, and limitation and future work will be dealt with in results and discussion part.

## 2    Method

In this section, we will explain problems when applying a model other than cellpose to challenge dataset. After that, preprocessing, detailed description of cellpose, and post processing will be explained. The information related to cellpose in this paper refer to the original cellpose paper[1].

### 2.1    Weekness of Existing Methods

Among the existing methods, simple methods used watershed.[8] The Watershed algorithm draws a topological map based on the grayscale value with threshold, and then separates the regions based on the intensity of the pixel. However, the watershed method is sensitive to noise and it becomes inaccurate when the boundaries are unclear because cells are attached to each other.In addition, if the size and shape of the cell change, some factor like thresholds must be modified[9]. Therefore, it is not suitable for a dataset of challenge, which generally has to work accurately on various cell images. Methods that distinguish background, cells, and boundaries, such as U-Net[2], model is more likely to predict that pixels are included in the boundaries if images contain high cell density and high boundary ratio. Therefore result is more likely to under segmented. Conversely, if some of the pixels contained in the boundaries of large cells are incorrectly predicted as cell, two or more cells are predicted as one large cell, adversely affecting the performance. Models using star-convex polygons such as Stardist[3] produce more than one star-convex polygons for elongated cells, so the mask does not fill the entire cell[10].

On the other hand, because cellpose is a neural net based on U-Net, there are skip connection and uses a style vector that encodes the style of the input. Therefore, it is more robust to noise and various cellular morphologies than methods using watershed. In addition to distinguishing cells and backgrounds per pixel, cellpose predicts the gradient of the flow toward the center of the cell, so there is more information to use compared to the inference method used in U-Net. Finally, unlike the stardist, the cellpose works well for elongated cells because it does not assumes the center and predict the distance to the outer boundary of the cell, but predicts the point where the flow gradient of each pixel converges as the center of the cell. Therefore, we chose the cellpose model for the cell segmentation task of the challenge.

### 2.2    Preprocessing

Since cellpose learns to predict the gradient of the flow, the mask of the original label must be converted to the gradient of the flow. That is, the image obtained by converting the original mask image according to the output format of the cell pose is a vector field. This vector field points to the center of the cell. However, if the cell's convexity is low, it is not necessary to point directly to the center of the cell. Instead, translating between pixels along the vector field was calculated to eventually converge to the center of the cell. Following the vector field inside the same cell converges at one point, so pixels can be said to be included in the same cell if output vector fields converge at same point.

The method to obtain the vector field from the mask uses the heat diffusion method. First, the midpoint of the vertical and horizontal boundaries of the cell is defined as the center point. However if the

central point is outside the cell, the nearest point inside cell is defined as the central point. The center point defined in this way is regarded as the source of heat, and the pixel value of the source is increased by 1 for each iteration. At the same time, each pixel inside the cell is updated to the average value of the 3x3 pixel region including itself. However, the outside of the cell is fixed to zero. The number of Iterations is determined to be twice the vertical and horizontal ranges to ensure that the heat is sufficiently diffused. The gradient of the energy distribution generated in this way is calculated to generate the final label to be used for training.

Meanwhile, in the input image, 1 to 99 percentiles of pixel intensity was converted from 0 to 1. In addition, images with more than three channels are discarded, leaving only two channels designated, and the second channel is filled with zero for the gray scale image. More details about channel will be explained in 2.3.2.

## 2.3 Cellpose

### 2.3.1 Model Structure and Loss Function

Basic structure of cellpose is U-Net which is like Figure 1. Downsampling pass and upsampling pass consists of four spatial scales. Each spatial scale consists of two residual block which is consists of two 3x3 convolution layers. In other words, one spatial scale contains four convolution layers. Each output of convolution layer perform batchnorm and ReLU operation. Not only there are skip connections in residual block, but also from down sampling pass to upsampling pass in same spatial scale. Operations that performed at each layer in downsampling pass is like 1.

$$
\begin{aligned}
\mathbf{x}'_t &= D_{2\times2}\left(\mathbf{x}_{t-1}\right) \\
\mathbf{x}^*_t &= F\left(F\left(\mathbf{x}'_t\right)\right) + P_{1\times1}\left(\mathbf{x}'_t\right) \\
\mathbf{x}_t &= F\left(F\left(\mathbf{x}^*_t\right)\right) + \mathbf{x}^*_t\,,
\end{aligned}
\tag{1}
$$

$D_{2\times2}$ is downsampling operation and $F$ is sequantial operation of convolution, batchnorm, and relu. Note that each $F$ has different parameters. What's more, $P_{1\times1}$ is 1x1convolution.

The last layer output of the downsampling pass is used to create a style vector. A style vector is a vector encoding a style of an input image. Each dimension of the style vector is the global average pooling per channel of the feature map which is the last output of the downsampling pass. Since the feature map has 256 channels, the style vector becomes a 256-dimensional vector. At this time, the density of cells may differ for each input, so the style vector is normalized. The style vector is delivered as an input to the residual block of each spatial scale of the upsampling pass.

The structure of the upsampling pass is opposite to downsampling pass in the order of the spatial scale, and the rest are the same. Instead, skip connection and style vector are added during the upsampling pass. The skip connection between the downsampling pass and the upsampling pass is connected by adding the feature map of the same spatial scale of the downsampling pass to the second convolution output of the corresponding upsampling pass. The Style vector is added to the remaining three outputs except for the first of the four convolution outputs of each residual block. Mathematically speaking this process is like expression2.

$$
\begin{aligned}
\mathbf{z}'_t &= U_{2\times2}\left(\mathbf{z}_{t+1}\right) \\
\mathbf{z}^*_t &= G\left(\mathbf{s}^*, F\left(\mathbf{z}'_t\right) + \mathbf{x_t}\right) + P_{1\times1}\left(\mathbf{z}'_t\right) \\
\mathbf{z}_t &= G\left(\mathbf{s}^*, G\left(\mathbf{s}^*, \mathbf{z}^*_t\right)\right) + \mathbf{z}^*_t\,,
\end{aligned}
\tag{2}
$$

$U_{2\times2}$ is upsampling operation and $G$ is operation that add style vector $s^*$ to feature map.

Input shape of cellpose is pre-processed 2channel microscopy image and the output is the probability of cell, horizontal, and vertical flow field for each pixel. Therefore last output of convolution in upsampling pass is 3channel map by 3channel 1x1 convolution operation. Horizontal flow and vertical flow is applied L2 loss because they are real number, and probability of cell is applied cross entropy loss $L_{CE}$ after sigmoid operation $\sigma$. Therefore, final loss function $L_{final}$ is like expression 3.

$$
L_{final} = \left\|\mathbf{y_0} - 5\mathbf{H}\right\|^2 + \left\|\mathbf{y_1} - 5\mathbf{V}\right\|^2 + L_{CE}\left(\sigma\left(\mathbf{y_2}\right), \mathbf{P}\right)
\tag{3}
$$

Where $\mathbf{y_0}, \mathbf{y_1}, \mathbf{y_3}$ means horizontal, vertical gradient and probability of cell. $\mathbf{H}, \mathbf{V}, \mathbf{P}$ is ground truth horizontal, vertical gradient and probability of cell.

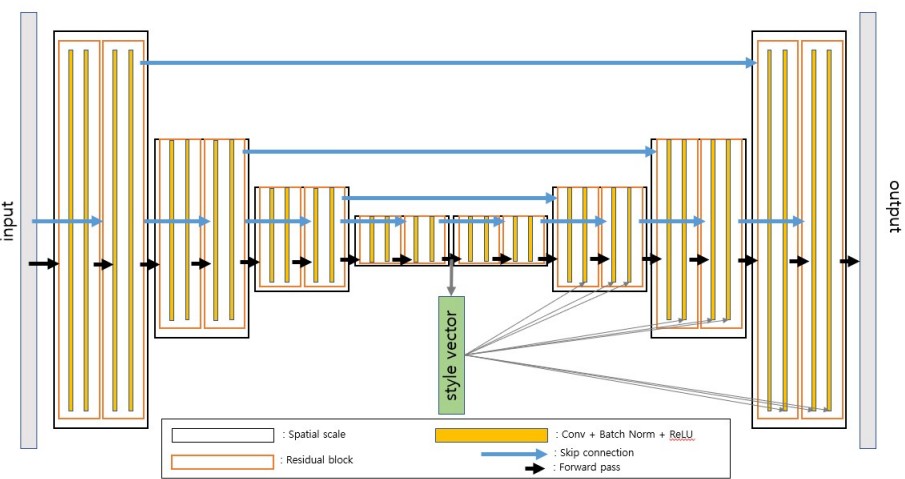

Figure 1: U-Net model used in cellpose.

### 2.3.2 Adapting Cellpose to Challenge Dataset

**Adapting channel**    Dataset of challenge consists of images obtained with various microscopy modalities. Therefore, the number of channel, file size , and file format of each modality are different. However, if you arbitrarily change the channel or modify the file format, the pretrained cellpose model may not be able to infer exactly. Therefore, we followed the method of processing the image in the original source code of the cellpose. The cellpose model we used is a cytoplasm model trained with a 2 channel image. Cytoplasm model was pretrained in the image, where the first channel is the channel to segment and the second channel is the optional channel representing the cell nucleus. However, in order to take advantage of the important colors of images in the dataset of challenge, such as dye, we used these images without changing them to gray scales. Therefore, like the original cellpose code, the gray scale image was made into a two-channel image with the second channel as zero, and the image with three or more channels was made into a two-channel image maintaining only designated two channel. In the given challenge dataset, we used green for the first channel and blue for the second channel because there are many green-stained cell images with the blue-stained cell nucleus.

**Size estimation**    Dataset of challenge consists of various resolutions. Therefore, even if modality and cell type are the same, there is a difference in the receptive field of the model, which may reduce accuracy. Cellpose solved this problem with the size estimation model. At test time, the downsampling pass of U-Net outputs a style vector from the input image. With a given style vector, the size estimation model guesses the diameter of the cells present in the image. After that, the ratio of the estimated diameter to the diameter trained by the model during training is calculated to resize the entire image. We retrained the size estimation model without using the cellpose pretrained model to deal with the fact that the dataset of challenge has various resolution. The size estimation model is created using a style vector generated by the training set after the U-Net model has been trained completely. The size estimation model is not a additional neural net, but a matrix A that satisfies the expression 4.

$$(XX^T + L)A = XY$$
$$where\ X = (S - \mu_S)^T,\ Y = (D - \mu_D) \tag{4}$$

$S$ represents style matrix made of style vector, and $D$ is matrix of diameter. $\mu_S$ and $\mu_D$ are style vector and diameter, respectively. $L = rI$ that $r$ is regularizer constant, $I$ is identity matrix which has same shape as S. In conclusion, size estimation model matrix A is a matrix that estimates diameter of cell by weighted sum of each dimension of style vector.

## 2.4 Post-processing

The output inferred by the model from the input image is a three-channel map having the same size as the input image. Each channel represents a cell probability, a horizontal, and a vertical gradient. Cellpose uses these three information to create a mask. It finds the center of the cell through gradient tracking, and pixels with a probability of cell 0.5 or more are the range of transfer along the gradient. When moving from pixels predicted to be cells to the neighboring pixel along the gradient, model predicts the converging point as the center of the cell and predicts that pixels converging to the same center belong to the same cell. Finally, a mask image is generated by allowing pixels belonging to the same cell to be included in one mask.

## 3 Experiments

### 3.1 Dataset

The cellpos we used were trained with various images. Author of cellpose mainly composed datasets with images obtained on the internet by searching keywords such as 'cytoplasm', 'cellular microscopy', and 'fluorescent cells'. The dataset consists of 361 images of fluorescently labeled protins on two DAPI-stained channels, 50 brightfield microscopy images, 58 membrane-labeled cells, 86 other microscopy images, and 98 non-microscopy images.

Dataset used in challenge consists of 300 bright field , 300 fluorescent , 200 phase-contrast , and 200 differential interference contrast. The cell types consist of red and white blood cells, plasma cells, hanseniaspora, and animal tissue cells, etc.

### 3.2 Implementation details

#### 3.2.1 Environment settings

The development environments and requirements are presented in Table 1.

Table 1: Development environments and requirements.

| System | Ubuntu 22.04.1 LTS |
|---|---|
| CPU | AMD Ryzen 7 5800X 8-Core Processor CPU@4.85GHz |
| RAM | 32GB; 2933 MT/s |
| GPU (number and type) | NVIDIA 1080Ti |
| CUDA version | 11.4 |
| Programming language | Python 3.9 |
| Deep learning framework | Pytorch (Torch 1.12.1, torchvision 0.2.2) |
| Specific dependencies | install cellpose |
| Code | `https://github.com/HyungjoByun/cvml_omnipose` |

#### 3.2.2 Training protocols

**Data augmentation** During training time, we flip images with random rotation and resize it to 224x224 size to train the U-Net model. To create a size estimation model, each image was passed through U-Net to create a style vector after randomly rotated, flipped, resized to 512x512.

The number of sampled style vector to create a style matrix $S$ is 10 times the size of the entire dataset. After sampling style vector, we can find $A$ from $S$ with linear regression. On the other hand, at test time, the image was divided into 224x224 tiles and inferred. In this case, the ratio of overlapping tiles is 10%.

**Hyper parameter settings** First, we initialize network with "LeCun" Uniform Initialization. We trained cellpose model 300 epochs using SGD optimizer with nesterov momentum ($\mu = 0.9$) and weight decay($\beta = 1e - 5$). Learning rate was set to 0.001, but it warmed up first 10epoch from 0 to 0.001, and after 210epoch, it is halved by 10epoch. The batch size was 8, and we resized images to 224×224, so patch size was 8×224×224. More details about training protocols are presented in Table 2

Table 2: Training protocols

| | |
|---|---|
| Network initialization | "LeCun" Uniform Initialization[11] |
| Batch size | 8 |
| Patch size | 8×224×224 |
| Total epochs | 300 |
| Optimizer | SGD with nesterov momentum ($\mu = 0.9$) and weight decay($\beta = 1e - 5$) |
| Initial learning rate (lr) | 0.001 |
| Lr decay schedule | epoch<10: start 0 and annealed linearly to 0.001, epoch>210: halved by 10epoch |
| Training time | 150min |
| Loss function | same as equation 3 |
| Number of model parameters | 6.6M |
| Number of flops | 15.7 G |

Table 3: Quantitative Results on tuning set

| Method | F1 Score |
|---|---|
| Baseline | 0.3112 |
| Baseline + Fine Tunning + Adaptive Segmentation Channel | 0.6387 |
| **Baseline + Fine Tunning + Adaptive Segmentation Channel + Diameter Estimation** | **0.7607** |

## 4    Results and discussion

### 4.1    Quantitative results on tuning set

In Table. 3, experiments were conducted in the supervised setting without using any unlabeled data. We used Cellpose backbone and chose the cyto2 model as a baseline. The baseline model did not show good performance on challenge dataset. As mentioned in the Method, we thought the reason was that the model did not adapt to data as diverse as the challenge dataset. In particular, the baseline model requires fine-tunning on a dataset of challenge, which has no special meaning for the channel, since baseline model have been trained to operate on images that each channel represents cell and nuclei. Therefore, we fine-tuned the baseline model to the challenge dataset. At this time, the channel was designated to use green and blue, taking advantage of the fact that dataset contains many images of green cells with blue nuclei. Images with three or more channels not containing green cell and blue nuclei were also use green and blue channels regardless of the definition of the channel. On gray scale images, we designated model to use single gray channel and channel representing nuclei was filled with zeros. These processes are considered as adaptive segmentation channel selection according to the image channel. More details about fine tuning is like section 3.2.2. As a result of fine-tuning, the F1 score increased by 0.3275 compared to baseline, resulting in the F1 score of 0.6387. Finally, our proposed method combined with the above methods and diameter estimation on each image. As a result, the F1 score was 0.7607, which was 0.4495 compared to Baseline, and 0.1220 higher than the Fine Tunning + Adaptive Segmentation Channel method.

### 4.2    Qualitative results on tuning set

**Effectiveness of Our Proposed Method** Fig. 2 shows the difference between the results of the cyto pretrained model and the proposed method. In the case of columns 1 and 2, it was observed that the proposed method was color-trained. In the case of column 3, as shown in (b), the pretrained cytoplasm model was not trained on case 35 at all, but our proposed method was trained on non-circular shapes as shown in (c). The result of column 3 shows that the cellpose model is robust for diverse shapes. Column 4 appears to be not recognized in (b), whereas in (c), the segmentation channel is recognized, and the inference is well established. In the case of columns 5 and 6, we observed that the recognition rate of our proposed method increased compared to the cytoplasm pretrained model.

**Effectiveness of Diameter Estimation** Fig. 3 shows the difference in the inference performance according to the diameter estimation. It has an image size of 640×480 for images of columns 1, for image of columns 2 has an image size of 2560×1920. In the absence of diameter estimation at a large image size, performance decreases significantly. The results of column 2 (b) show that cells are not recognized when the inference was performed, and the recognition rate increases when the

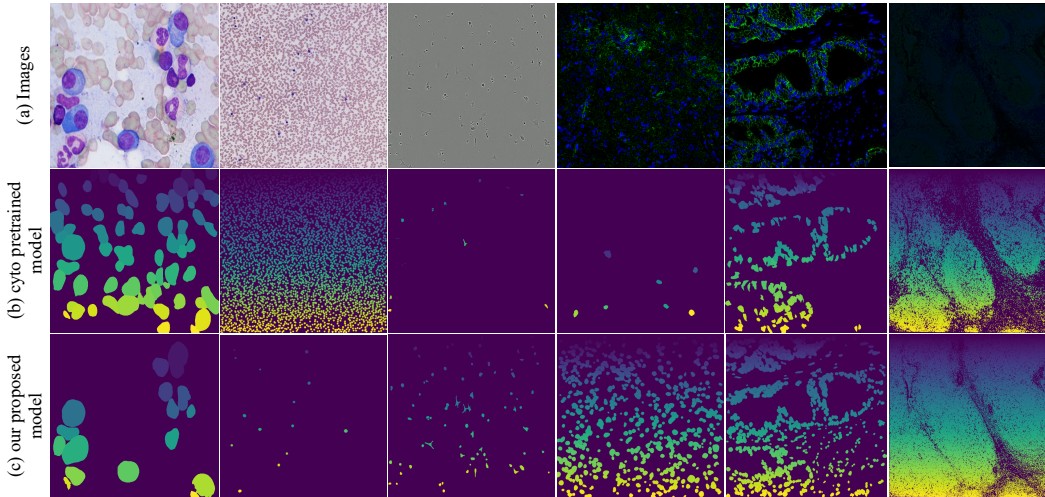

Figure 2: Qualitative results of proposed method. (a) Input images, case 1, 4, 35, 49, 67, 101 from the left (b) Inference results on cytoplasm pretrained model. (c) Inference results of our proposed method from the left

diameter estimation is performed. The model performed diameter estimation with 58.32 and 125.02 for columns 1 and 2, indicating that performance deteriorates when there is a large difference in mean diameter values.

Looking at column 4 (b), it is expected that the difference between the estimated average diameter and the default setting will be large in a state where the cell is not recognized at all. The predicted mean diameter values in columns 3 and 4 are 52.52 and 52.30, which are greater than 30, but are similar to the predicted mean diameter values in column 1. However, although the average diameter of the first row is similar, the inference is performed well. From this, it is presumed that the estimated diameter value is measured differently from the average diameter value of the cell when the image size of columns 3 and 4 is adjusted. Therefore, if the diameter estimation is performed separately for the image, the inference seems to work well.

When diameter estimation was performed through these qualitative results, it was observed that the recognition rate of cells increased for various image sizes. Our proposed method performed finetunning, adaptive segmentation channel, and diameter estimation, and the F1 score at this time was 0.7607. On the other hand, when the diameter estimation was not performed, the F1 score was 0.6387, indicating a performance difference of 0.1220.

**Effectiveness of flow threshold** In order to find out the difference in the recognition rate of the colonial morphology according to the flow threshold value, an experiment was conducted with the flow threshold set to 0. The results of the cytoplasm pretrained model for cases 74, 77, and 78 and the results according to the flow threshold value. It was confirmed that the colonial morphology was recognized to some extent in the cytoplasm pretrained model, but the image of each cell was not grasped. However, when the flow threshold is set to 0.4, the overall recognition rate decreases, and when the flow threshold is 0, the recognition rate increases. Compared to the case where the flow threshold is set to 0, many recognized cells disappear when the flow threshold is set to 0.4. This means that the model inferred uncertainly and inferred differently from the real cell morphology. Contrary to the prediction that performance would have improved due to a higher recognition rate, in Table. 4, it was found that there was a performance drop of 0.0523 with 0.7607 and 0.7084 respectively when the flow threshold was designated as 0.

The identified problems were very small in the data distribution of these labeled bacilli training images, only 0.3% in the provided dataset. This means that training will not go well in this image style. In addition, the bacillus shape is recognized as uncertained because it is different from the shape of the cell, which accounts for a significant portion of the dataset distribution.

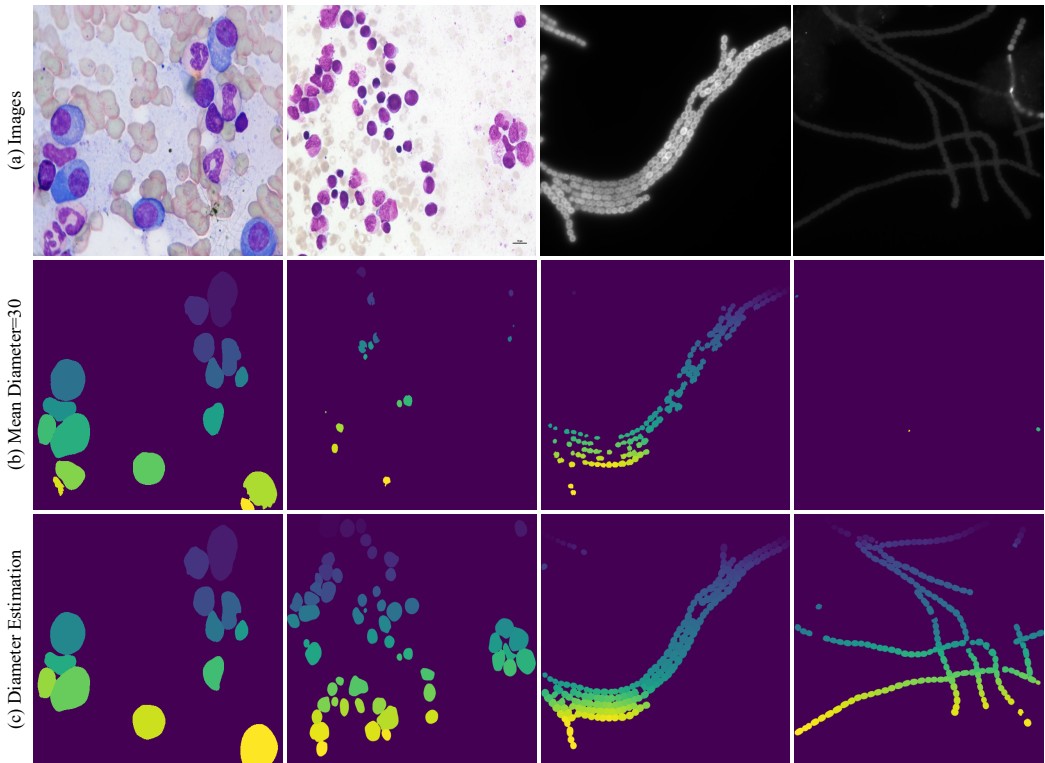

Figure 3: Qualitative results for the effects of diameter estimation. (a) Input images. From the left to the interference results in cases 1, 2, 72, 73 (b) Mean Diameter=30. (c) Diameter estimation for each image. Proposed method setting.

### 4.3 Limitation and future work

**Limitation** Cell types with a small absolute number in the labeled dataset are not well recognized. In particular, this phenomenon was found for images having a bacillus shape as cases 74, 77, and 78. If the number of labeled data is small, the real shape of a specific cell is perceived as uncertain. Also, since there is no image for the case in the unlabeled data, even if the unlabeled data is used, the recognition rate for the corresponding image type may not be raised. In addition, cellpose backbone uploads all images to RAM during pre-processing, but we were unable to conduct experiments with our computer specifications due to capacity issues in the pre-processing process in the process of using unlabeled data.

**Future work** We will apply semi-supervised learning through unlabeled data through consistency regularization that maintains prediction cosistency by giving pertubation to unlabeled data. Also, we will solve the capacity issue in the pre-processing process of cellpose and conduct an experiment using unlabeled data. In addition, through optimization of the flow threshold value, we will create a model that recognizes close to the real shape of the cell, aiming to improve the recognition rate for the relevant cases 74, 77 and 78.

## 5 Conclusion

Our proposed method was optimized through various hyper parameter tuning provided by cellpose. In particular, by applying diameter estimation to each cell image, the performance through flow threshold tuning was discovered. Our experimental results show that a higher degree of accuracy can be achieved for the data provided than the cyto pretrained model. For the validation(tuning) set, the F1 score of the cyto pretrained model was 0.3112, and our proposed method showed an F1 score of 0.7607, which improved the performance of 0.4495.

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

# A  Discussion

Table 4: Quantitative Results on tuning set

| Diameter Estimation | Flow Threshold=0 | Merge Channel | Style Vector=10 | Resize=(512,512) | F1 Score |
|:---:|:---:|:---:|:---:|:---:|:---:|
| | | | | | 0.6387 |
| ✓ | | | | | **0.7607** |
| ✓ | ✓ | | | | 0.7084 |
| ✓ | | ✓ | | | 0.7336 |
| ✓ | | | ✓ | | 0.6805 |
| ✓ | | | | ✓ | 0.6518 |

As shown in Table. 4, when the diameter estimation = 30, the performance degraded 0.1220 compared to the no diameter estimation setting which F1 score was 0.7607. Therefore, a subsequent experiment was conducted with the diameter estimation applied as default. Setting the flow threshold=0 increases the cell recognition rate, but the model infers differently than the actual cell morphology.

We also conducted an experiment to merging channels. Merging channels means that the pixel values of the remaining channels, excluding the already chosen green and blue channels, are added to the

green and blue channels and divided by the combined number of channels. Mathematically it is like Expression 5.

$$I_k = \frac{1}{N_C - 1} \left( I_k + \sum_{t \in C'} I_t \right)$$
$$where \; k \in \{g, b\}, C' = C - \{g, b\} \tag{5}$$

In this case, $I_k$ is a pixel value of the channel k. $g,b$ means channels corresponding to green and blue in an image with RGB channels, and $C$ is a set of channels in the image. $N_C$ is the number of channels. If image had one channel, unlike the conventional method of filling the second channel with zero, the image were copied into two and assigned one as the second channel. As a result of the experiment of merging channels, the performance was reduced compared to the method discarding the unselected channel resulting in F1 score 0.7336.

In the cytoplasm pretrained model, the style vector was fixed to 1 and training proceeded. When directly observing the dataset, the data can be classified into 10 types, so we set style vector=10 before training. In this case, it was estimated that the performance decreased significantly because there were too many clusters to train. Finally, an experiment was performed to resize the training image to $512 \times 512$. As a result, the F1 score decreased by 0.1089 to 0.6518 compared to the existing $224 \times 224$ size with an F1 score of 0.7607.

