# OpenReview forum: "Cell Segmentation in Multi-modality High-Resolution Microscopy Images with Cellpose"
_NeurIPS.cc/2022/Challenge/CellSeg — Submitted to NeurIPS CellSeg 2022_

### Official Review · Reviewer_77Zb · 2022-12-19
**Paper Review**

**Rating:** 4
**Confidence:** 4

**Review:**

# Summary
This paper employed and modified the Cellpose method for the cell segmentation task from images. The authors designated the model in terms of input channels and added the size estimation for better prediction. The background and method details were demostrated clearly.

# Pros
1. The modification of input channels adapted the Cellpose method for any image types, which is suitable for the competition.
2. The size estimation part helps the model to better predict cells in different resolutions.

# Cons
1. The false positive (FP) is an important issue in segmentation task. I suggested the authors further evaluated their method in terms of FP.
2. Although stated in the future work part, the paper did not proposed a way to use unlabeled data to perform semi-supervised learning, which is likely to improve the performance.
3. Besides the albation experiment, the authors should compare their method with other segmentation models.

---

### Official Review · Reviewer_D3JW · 2022-12-26
**paper well structured, more novelty needed**

**Rating:** 6
**Confidence:** 4

**Review:**

# Overall
Generally the paper contains all required information to reproduce similar results and is well structured. A modified cellpose model has been demonstrated to perform instance segmentation on dataset having various characteristics. In detail, channel has been adapted and a size estimation model has also been proposed. The final results have also confirmed the technical soundness.

# Semi-supervised learning
According to the description, it seems that semi-supervised learning has not been utilized, which could possibly improve the final performance.

# Speed
Is there any info about the inference speed? According to my understanding, original cellpose is not very fast. Since there are many large images in this competition, are there any specific strategies applied to the modified model to accelerate the speed?

# Summary
The theoretical analysis of size estimation is good written, and detailed ablations have also been demonstrated, but more novelty could be added, e.g. semi-supervised learning or other training/inference tricks.

---

### Official Review · Reviewer_uqwc · 2023-01-08
**Little contribution to multi-modality cell segmentation**

**Rating:** 4
**Confidence:** 4

**Review:**

The paper shows a framework that adopts Cellpose for multi-modality cell segmentation challenge. Compared to the original Cellpose, the proposed method considers different image channels and various image resolutions. Even though the paper gives a relative complete method and experimental report, the contribution of the paper is not significant, and the modification on baseline Cellpose is small.

Besides, there are other three major concerns:

1. The claims of the related work are not very accurate. For example, in Sec. I Introduction, it claims that Cellpose works better than Stardist in elongated and complex shapes of cell because no predetermined polygon is assumed.  The reviewer does not agree with the point. Stardist can also model complex shape of cells using high-density rays.

In Sec.2 method, it discusses the disadvantages of the watershed method. However, it ignores the discussion of the efficiency of watershed as a post-processing of deep learning models.

2. Details of the proposed size estimation is missing. Size estimation is an important part of the proposed method. However, it only utilizes a few sentences to illustrate the size estimation method. The details of how the size estimation is applied is missing. And in function (4), it doesn't show what is style matrix and what is diameter matrix, how the regularizer constant is set.

3. The comparison experiments are not complete.  No other SOTA methods are compared.

---

### Official Review · Reviewer_RxuP · 2023-01-09
**A complete solution but limited contributions**

**Rating:** 4
**Confidence:** 3

**Review:**

Summary:
The authors implement a Cellpose model on the challenge dataset, by adapting the input channels of different image modalities and adaptively resizing the input sizes with a size estimation method.

Pros:
1. The experimental results show the effectiveness of the size estimation approach.

Cons:
1. It is unclear why the misalignment of channels is a critical difficulty. What about using 3 channels (RGB) and training/pretraining the Cellpose model from scratch?
2. The original paper of Cellpose also use a size estimation module. What is new in the size estimation part of this draft? This draft looks a re-implementation of Cellpose.

---

### Decision · Program_Chairs · 2023-01-19

Accept